# Identification of Conditional Causal Effects under Markov Equivalence

**Amin Jaber**
Purdue University
jaber0@purdue.edu

**Jiji Zhang**
Lingnan University
jijizhang@ln.edu.hk

**Elias Bareinboim**
Columbia University
eb@cs.columbia.edu

## Abstract

Causal identification is the problem of deciding whether a post-interventional distribution is computable from a combination of qualitative knowledge about the data-generating process, which is encoded in a causal diagram, and an observational distribution. A generalization of this problem restricts the qualitative knowledge to a class of Markov equivalent causal diagrams, which, unlike a single, fully-specified causal diagram, can be inferred from the observational distribution. Recent work by (Jaber et al., 2019a) devised a complete algorithm for the identification of *unconditional* causal effects given a Markov equivalence class of causal diagrams. However, there are identifiable *conditional* causal effects that cannot be handled by that algorithm. In this work, we derive an algorithm to identify conditional effects, which are particularly useful for evaluating conditional plans or policies.

## 1  Introduction

The graphical approach to causal inference is becoming an important tool for assessing the efficacy of actions or policies (Pearl, 2000; Bareinboim and Pearl, 2016). In this approach, data from an observational probability distribution $P$ is associated with a causal diagram (e.g., Fig. 1a) in which nodes correspond to measured variables, directed edges represent direct causal relations, and bi-directed edges encode spurious associations due to unmeasured confounding variables. Performing an *action* $\mathrm{do}(\mathbf{X} = \mathbf{x})$ eliminates the impact of other variables on those in $\mathbf{X}$ by fixing the values of the latter and induces an interventional distribution, denoted $P_{\mathbf{x}}$. Whether, and if so how, aspects of $P_{\mathbf{x}}$ can be determined from the observational distribution together with the causal diagram is known as the problem of causal identification.

In this work, we focus on *conditional* causal effects, of the form $P_{\mathbf{x}}(\mathbf{y}|\mathbf{z})$, which denotes the conditional probability of $\mathbf{Y} = \mathbf{y}$ given $\mathbf{Z} = \mathbf{z}$ according to the interventional distribution $P_{\mathbf{x}}$. Such conditional effects are particularly useful when what is at stake is the consequence of conditional plans or policies, in which what value or probability distribution to impose on $\mathbf{X}$ is contingent on the value of $\mathbf{Z}$ (Pearl and Robins, 1995). When the available knowledge is sufficient to delineate the causal diagram, a number of criteria, including a complete algorithm, for identifying conditional effects are known (Pearl, 1995; Spirtes et al., 2000; Tian, 2004; Shpitser and Pearl, 2006). However, we are usually in a position where background knowledge is not nearly enough to give us confidence on a single causal diagram. In such situations, forcing a single diagram easily leads to false modeling assumptions and, consequently, misleading inferences.

Instead of specifying the causal diagram based on expert knowledge, one may adopt a more data-driven approach and attempt to learn it from data. However, from observational data, it is common that only a Markov equivalence class of causal diagrams can be consistently estimated (Verma, 1993; Spirtes et al., 2001; Zhang, 2008b). A distinguished characterization of the Markov equivalence class uses *partial ancestral graphs (PAGs)*. Fig. 1b shows the PAG learnable

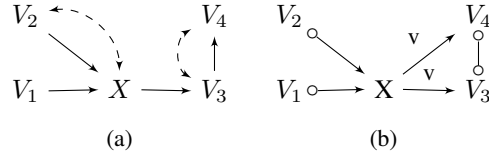

Figure 1: Causal diagram (left) and the inferred PAG (right).

from observational data that is consistent with the causal diagram depicted in Fig. 1a. The directed edges in a PAG represent causal relations (that are not necessarily direct) and the circle marks stand for structural uncertainty. Labeled edges (with v) signify the absence of unmeasured confounders.

In this work, we study the problem of using invariant structural features in a Markov equivalence class (learnable from observational data) to identify conditional causal effects. Identification from an equivalence class is considerably more challenging than from a single diagram due to the structural uncertainties. Zhang (2007) extended Pearl's do-calculus to PAGs. However, it is computationally hard to decide whether there exists (and, if so, to find) a sequence of derivations in the generalized calculus to identify the effect of interest. More recently, a complete algorithm was devised for identifying unconditional causal effects given a PAG (Jaber et al., 2019a). This algorithm can be used to identify conditional effects of the form $P_{\mathbf{x}}(\mathbf{y}|\mathbf{z})$ whenever the joint effect $P_{\mathbf{x}}(\mathbf{y}, \mathbf{z})$ is identifiable. However, as we will show, many conditional effects are identifiable while the corresponding joint effect is not.[1] Specifically, we make the following contributions:

1. We establish a novel decomposition that serves to reduce a targeted conditional causal distribution into components that are easier to identify.

2. Based on the decomposition, we develop an algorithm to compute the effect of an arbitrary set of intervention variables on an arbitrary outcome set while conditioning on a third disjoint set, from a PAG and an observational distribution. We show that this algorithm subsumes that of (Jaber et al., 2019a).

## 2 Preliminaries

In this section, we introduce the basic setup and notations. Boldface capital letters denote sets of variables, while boldface lowercase letters stand for value assignments to those variables.

**Structural Causal Models.** We use Structural Causal Models (SCMs) (Pearl, 2000, pp. 204-207) as our basic semantical framework. Formally, an SCM $M$ is a 4-tuple $\langle \mathbf{U}, \mathbf{V}, \mathbf{F}, P(\mathbf{U}) \rangle$, where $\mathbf{U}$ is a set of exogenous (latent) variables and $\mathbf{V}$ is a set of endogenous (measured) variables. $\mathbf{F}$ represents a collection of functions $\mathbf{F} = \{f_i\}$ such that each endogenous variable $V_i \in \mathbf{V}$ is determined by a function $f_i \in \mathbf{F}$, where $f_i$ is a mapping from the respective domain of $\mathbf{U}_i \cup \mathbf{Pa}_i$ to $V_i$, $\mathbf{U}_i \subseteq \mathbf{U}$, $\mathbf{Pa}_i \subseteq \mathbf{V} \setminus V_i$. The uncertainty is encoded through a probability distribution over the exogenous variables, $P(\mathbf{U})$. Every SCM is associated with one causal diagram where every variable $\mathbf{V} \cup \mathbf{U}$ is a node, and an arrow is drawn from each member of $\mathbf{U}_i \cup \mathbf{Pa}_i$ to $V_i$. Following standard practice, when drawing a causal diagram, we omit the exogenous nodes and add a bi-directed arc between two endogenous nodes if they share an exogenous parent. We restrict our study to recursive systems, which means that the corresponding diagram will be acyclic. The marginal distribution induced over the endogenous variables $P(\mathbf{V})$ is called observational, and factorizes according to the causal diagram, i.e.:

$$P(\mathbf{v}) = \sum_{\mathbf{u}} \prod_i P(v_i|\mathbf{pa}_i, \mathbf{u}_i)P(\mathbf{u}) \tag{1}$$

Within the structural semantics, performing an action $X = x$ is represented through the do-operator, $do(X = x)$, which encodes the operation of replacing the original equation for $X$ by the constant $x$ and induces a submodel $M_x$. The resulting distribution is denoted by $P_x$, which is the main target for identification in this paper. For details on structural models, we refer readers to (Pearl, 2000).

**Ancestral Graphs.** We now introduce a graphical representation of equivalence classes of causal diagrams. A *mixed* graph can contain directed and bi-directed edges. $A$ is an ancestor of $B$ if there is a directed path from $A$ to $B$. $A$ is a *spouse* of $B$ if $A \leftrightarrow B$ is present. An *almost directed cycle* happens when $A$ is both a spouse and an ancestor of $B$. An *inducing path* is a path on which every node (except for the endpoints) is a collider on the path (i.e., both edges incident to the node are into it) and is an ancestor of an endpoint of the path. A mixed graph is *ancestral* if it does not contain directed or almost directed cycles. It is *maximal* if there is no inducing path between any two non-adjacent nodes. A *Maximal Ancestral Graph* (MAG) is a graph that is both ancestral and maximal (Richardson and Spirtes, 2002).

In general, a causal MAG represents a set of causal diagrams with the same set of observed variables that entail the same conditional independence and ancestral relations among the observed variables. Different MAGs may be Markov equivalent in that they entail the exact same independence model. A partial ancestral graph (PAG) represents an equivalence class of MAGs $[\mathcal{M}]$, which shares the same adjacencies as every MAG in $[\mathcal{M}]$ and displays all and only the invariant edge marks (i.e., edge marks that are shared by all members of $[\mathcal{M}]$). A circle indicates an edge mark that is not invariant.

A PAG is learnable from the independence model over the observed variables, and the FCI algorithm is a standard method to learn such an object (Zhang, 2008b). In short, a PAG represents a class of causal diagrams with the same observed variables that entail the same independence model over the observed variables.

**Graphical Notions.** Given a causal diagram, a MAG, or a PAG, a path between $X$ and $Y$ is *potentially directed (causal)* from $X$ to $Y$ if there is no arrowhead on the path pointing towards $X$. $Y$ is called a *possible descendant* of $X$ and $X$ a *possible ancestor* of $Y$ if there is a potentially directed path from $X$ to $Y$. $Y$ is called a *possible child* of $X$ and $X$ a *possible parent* of $Y$ if they are adjacent and the edge is not into $X$. For a set of nodes $\mathbf{X}$, let $\mathtt{Pa}(\mathbf{X})$ ($\mathtt{Ch}(\mathbf{X})$) denote the union of $\mathbf{X}$ and the set of possible parents (children) of $\mathbf{X}$, and let $\mathtt{An}(\mathbf{X})$ denote the union of $\mathbf{X}$ and the set of possible ancestors of $\mathbf{X}$. Let $\mathtt{Pa}^*(\mathbf{X})$ denote $\mathtt{Pa}(\mathbf{X})$ excluding the possible parents of $\mathbf{X}$ due to circle edges ($\circ\!-\!\circ$). Similarly, $\mathtt{Ch}^*(\mathbf{X})$ denotes $\mathtt{Ch}(\mathbf{X})$ excluding the possible children of $\mathbf{X}$ due to circle edges. For convenience, we use an asterisk (*) as a wildcard to denote any possible mark of a PAG ($\circ, >, -$) or a MAG ($>, -$). If the edge marks on a path between $X$ and $Y$ are all circles, we call the path a *circle path*. We refer to the closure of nodes connected with circle paths as a *bucket*. Obviously, given a PAG, nodes are partitioned into a unique set of buckets.

A directed edge $X \rightarrow Y$ in a MAG or a PAG is *visible* if there exists no causal diagram in the corresponding equivalence class where there is an inducing path between $X$ and $Y$ that is into $X$. This implies that a visible edge is not confounded ($X \leftarrow\!-\!\rightarrow Y$ doesn't exist). Which directed edges are visible is easily decidable by a graphical condition (Zhang, 2008a), so we simply mark visible edges by $v$. For brevity, we refer to any edge that is not a visible directed edge as *invisible*.

**Identification in Causal Diagrams.** Tian and Pearl (2002) introduced a decomposition of a causal diagram into a set of so-called *c-components* (confounded components).

**Definition 1** (C-Component). *In a causal diagram, two nodes are said to be in the same c-component iff they are connected by a bi-directed path, i.e., a path composed solely of bi-directed edges.*

The significance of c-components and their decomposition is evident from (Tian, 2004, Lemmas 2, 3), which are the basis for the proposed algorithm for identifying conditional causal effects. For any set $\mathbf{C} \subseteq \mathbf{V}$, $Q[\mathbf{C}]$ denotes the post-intervention distribution of $\mathbf{C}$ under an intervention on $\mathbf{V} \setminus \mathbf{C}$.

$$Q[\mathbf{C}] = P_{\mathbf{v} \setminus \mathbf{c}}(\mathbf{c}) = \sum_{\mathbf{u}} \prod_{\{i \mid V_i \in \mathbf{C}\}} P(v_i | \mathbf{pa}_i, \mathbf{u}_i) P(\mathbf{u}) \tag{2}$$

Obviously, $Q[\mathbf{C}]$ functionally depends on $\mathbf{C}$ and the corresponding parents, i.e., $\mathtt{Pa}(\mathbf{C})$. Moreover, $Q[\mathbf{C}]$ decomposes into a product of sub-queries over the c-components in $\mathcal{D}_{\mathbf{C}}$, the induced subgraph of the causal diagram $\mathcal{D}$ over $\mathbf{C}$. That is, $Q[\mathbf{C}] = \prod_i Q[\mathbf{C}_i]$, where $\mathbf{C}_i$ is a c-component in $\mathcal{D}_{\mathbf{C}}$.

## 3 Unconditional Causal Effect

In this section, we review the techniques developed in (Jaber et al., 2019a) for identifying uncondi-tional causal effects. The notion of pc-component (Def. 2) in MAGs and PAGs generalizes that of

**Algorithm 1** IDP($\mathbf{x}, \mathbf{y}$) given PAG $\mathcal{P}$
___
    **Input:** two disjoint sets $\mathbf{X}, \mathbf{Y} \subset \mathbf{V}$
    **Output:** Expression for $P_\mathbf{x}(\mathbf{y})$ or `FAIL`

 1: Let $\mathbf{D} = \text{An}(\mathbf{Y})_{\mathcal{P}_{\mathbf{V} \setminus \mathbf{x}}}$
 2: $P_\mathbf{x}(\mathbf{y}) = \sum_{\mathbf{d} \setminus \mathbf{y}} \text{IDENTIFY}(\mathbf{D}, \mathbf{V}, P)$

 3: **function** IDENTIFY($\mathbf{C}, \mathbf{T}, Q = Q[\mathbf{T}]$)
 4:    **if** $\mathbf{C} = \emptyset$ **then return** 1
 5:    **if** $\mathbf{C} = \mathbf{T}$ **then return** $Q$
    /* In $\mathcal{P}_\mathbf{T}$, let $\mathbf{B}$ denote a bucket, and let $C^\mathbf{B}$ denote the pc-component of $\mathbf{B}$ */
 6:    **if** $\exists \mathbf{B} \subset \mathbf{T} \setminus \mathbf{C}$ such that $C^\mathbf{B} \cap \text{Ch}(\mathbf{B}) \subseteq \mathbf{B}$ **then**
 7:        Compute $Q[\mathbf{T} \setminus \mathbf{B}]$ from $Q$;                          ▷ via Proposition 2
 8:        **return** IDENTIFY($\mathbf{C}, \mathbf{T} \setminus \mathbf{B}, Q[\mathbf{T} \setminus \mathbf{B}]$)
 9:    **else if** $\exists \mathbf{B} \subset \mathbf{C}$ such that $\mathcal{R}_\mathbf{B} \neq \mathbf{C}$ **then**
10:        **return** $\frac{\text{IDENTIFY}(\mathcal{R}_\mathbf{B}, \mathbf{T}, Q) \cdot \text{IDENTIFY}(\mathcal{R}_{\mathbf{C} \setminus \mathcal{R}_\mathbf{B}}, \mathbf{T}, Q)}{\text{IDENTIFY}(\mathcal{R}_\mathbf{B} \cap \mathcal{R}_{\mathbf{C} \setminus \mathcal{R}_\mathbf{B}}, \mathbf{T}, Q)}$        ▷ by Proposition 3
11:    **else**
12:        **throw** `FAIL`
___

c-component in a causal diagram. Being in the same pc-component is a necessary condition for two nodes to be in the same c-component in some causal diagram in the corresponding equivalence class (Prop. 1). As a special case of Def. 2, two nodes are in the same *definite c-component* (*dc-component*) if they are connected with a bi-directed path, i.e., a path composed solely of bi-directed edges.

**Definition 2** (PC-Component). *In a MAG, a PAG, or any induced subgraph thereof, two nodes are in the same possible c-component (pc-component) if there is a path between them such that (1) all non-endpoint nodes along the path are colliders, and (2) none of the edges is visible.*

**Proposition 1.** *Let $\mathcal{P}$ be a MAG or a PAG over $\mathbf{V}$, and $\mathcal{D}$ be any causal diagram in the equivalence class represented by $\mathcal{P}$. For any $X, Y \in \mathbf{A} \subseteq \mathbf{V}$, if $X$ and $Y$ are in the same c-component in $\mathcal{D}_\mathbf{A}$, then $X$ and $Y$ are in the same pc-component in $\mathcal{P}_\mathbf{A}$.*

Using the above notions, the following identification criterion is derived where the intervention is on a bucket rather than a single node and the input distribution is possibly interventional. The expression depends on a partial topological order (*PTO*) over the nodes, which is a topological order over the buckets. A detailed discussion can be found in (Jaber et al., 2018).

**Proposition 2.** *Let $\mathcal{P}$ denote a PAG over $\mathbf{V}$, $\mathbf{T}$ be a union of a subset of the buckets in $\mathcal{P}$, and $\mathbf{X} \subset \mathbf{T}$ be a bucket. Given $P_{\mathbf{v} \setminus \mathbf{t}}$ (i.e., $Q[\mathbf{T}]$), and a partial topological order $\mathbf{B}_1 < \cdots < \mathbf{B}_m$ with respect to $\mathcal{P}_\mathbf{T}$, $Q[\mathbf{T} \setminus \mathbf{X}]$ is identifiable if and only if, in $\mathcal{P}_\mathbf{T}$, there does not exist $Z \in \mathbf{X}$ such that $Z$ has a possible child $C \notin \mathbf{X}$ that is in the pc-component of $Z$. If identifiable, then the expression is given by*

$$Q[\mathbf{T} \setminus \mathbf{X}] = \frac{P_{\mathbf{v} \setminus \mathbf{t}}}{\prod_{\{i | \mathbf{B}_i \subseteq S^\mathbf{X}\}} P_{\mathbf{v} \setminus \mathbf{t}}(\mathbf{B}_i | \mathbf{B}^{(\mathbf{i-1})})} \times \sum_\mathbf{x} \prod_{\{i | \mathbf{B}_i \subseteq S^\mathbf{X}\}} P_{\mathbf{v} \setminus \mathbf{t}}(\mathbf{B}_i | \mathbf{B}^{(\mathbf{i-1})}),$$

*where $S^\mathbf{X} = \bigcup_{Z \in \mathbf{X}} S^Z$, $S^Z$ being the dc-component of $Z$ in $\mathcal{P}_\mathbf{T}$, and $\mathbf{B}^{(i-1)}$ denoting the set of nodes preceding bucket $\mathbf{B}_i$ in the partial order.*

For example, given the PAG in Fig. 1b, $X$ is not in the same pc-component with any of its possible children $V_3, V_4$, hence $P_x(v_1, \ldots, v_4)$ is computable from the observational distribution $P(\mathbf{v})$. Another important result is decomposing a target quantity $Q[\mathbf{C}]$ into a product of smaller quantities. Such a decomposition is obtained in Proposition 3 using the *Region* construct (Def. 3).

**Definition 3** (Region $\mathcal{R}_\mathbf{A}^\mathbf{C}$). *Given a PAG or a MAG $\mathcal{P}$ over $\mathbf{V}$, and $\mathbf{A} \subseteq \mathbf{C} \subseteq \mathbf{V}$. Let the region of $\mathbf{A}$ with respect to $\mathbf{C}$, denoted $\mathcal{R}_\mathbf{A}^\mathbf{C}$, be the union of the buckets that contain nodes in the pc-component of $\mathbf{A}$ in the induced subgraph $\mathcal{P}_\mathbf{C}$.*

**Proposition 3.** *Given a PAG $\mathcal{P}$ over $\mathbf{V}$ and set $\mathbf{C} \subseteq \mathbf{V}$, $Q[\mathbf{C}]$ can be decomposed as follows.*

$$Q[\mathbf{C}] = \frac{Q[\mathcal{R}_\mathbf{A}].Q[\mathcal{R}_{\mathbf{C} \setminus \mathcal{R}_\mathbf{A}}]}{Q[\mathcal{R}_\mathbf{A} \cap \mathcal{R}_{\mathbf{C} \setminus \mathcal{R}_\mathbf{A}}]}$$

*where $\mathbf{A} \subset \mathbf{C}$ and $\mathcal{R}_{(.)} = \mathcal{R}_{(.)}^\mathbf{C}$.*

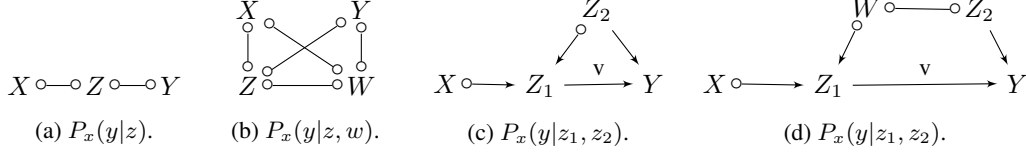

Figure 2: Sample PAGs with identifiable conditional causal effects.

Propositions 2 and 3 are utilized in Algorithm 1 which is sound and complete for identifying unconditional causal effects given a PAG (Jaber et al., 2019a).

## 4 Conditional Causal Effects

We formalize the notion of identifiability from a PAG using the following definition, which generalizes the causal-diagram-specific notion (Tian, 2004).

**Definition 4** (Causal-Effect Identifiability). *The causal effect of a set of variables $\mathbf{X}$ on a disjoint set of variables $\mathbf{Y}$ conditioned on another set $\mathbf{Z}$ is said to be identifiable from a PAG $\mathcal{P}$ if the quantity $P_{\mathbf{x}}(\mathbf{y}|\mathbf{z})$ can be computed uniquely from the observational distribution $P(\mathbf{V})$ given every causal diagram $\mathcal{D}$ (represented by a MAG) in the Markov equivalence class represented by $\mathcal{P}$.*

Given a PAG $\mathcal{P}$ and a conditional causal effect $P_{\mathbf{x}}(\mathbf{y}|\mathbf{z})$, we can rewrite the quantity as follows. Hence, if $P_{\mathbf{x}}(\mathbf{y}, \mathbf{z})$ is identifiable, then $P_{\mathbf{x}}(\mathbf{y}|\mathbf{z})$ is identifiable as well.

$$P_{\mathbf{x}}(\mathbf{y}|\mathbf{z}) = \frac{P_{\mathbf{x}}(\mathbf{y}, \mathbf{z})}{\sum_{\mathbf{y}} P_{\mathbf{x}}(\mathbf{y}, \mathbf{z})}$$

For example, $P_{z_1}(y, z_2)$ is identifiable in the PAG of Figure 2c with the following (simplified) expression via Algorithm 1. Hence, both $P_{z_1}(y|z_2)$ and $P_{z_1}(z_2|y)$ are identifiable.

$$P_{z_1}(y, z_2) = Q[Y, Z_2] = P(y|z_1, z_2)P(z_2)$$

However, not all identifiable conditional effects can be identified this way. Consider the PAG in Fig. 2d and the conditional effect $P_x(y|z_1, z_2)$. Whereas $P_x(y, z_1, z_2)$ is not identifiable by Algorithm 1 and hence the conditional effect is not identifiable *simpliciter*, $P_x(y|z_1, z_2)$ turns out to be identifiable as we show later. Therefore, Algorithm 1, though complete for identifying unconditional effects, is unable to compute many identifiable conditional effects.

To do better, we start by generalizing the notion of $Q[\cdot]$ to accommodate conditioning.

**Definition 5.** *For any pair of disjoint sets $\mathbf{C}, \mathbf{Z} \subseteq \mathbf{V}$, we define the quantity $Q[\mathbf{C}|\mathbf{Z}]$, given below, to be the post-intervention distribution of $\mathbf{C}$ conditional on $\mathbf{Z}$ under an intervention on $\mathbf{V} \setminus (\mathbf{C} \cup \mathbf{Z})$.*

$$Q[\mathbf{C}|\mathbf{Z}] = \frac{Q[\mathbf{C} \cup \mathbf{Z}]}{\sum_{\mathbf{c}} Q[\mathbf{C} \cup \mathbf{Z}]}$$

In what follows, we utilize Definition 5 to derive an algorithm for conditional causal effect identification. The following proposition shows a way to rewrite a given conditional effect in terms of the notion in Definition 5.[2]

**Proposition 4.** *Given distribution $P(\mathbf{V})$, causal PAG $\mathcal{P}$ over $\mathbf{V}$, and target effect $P_{\mathbf{x}}(\mathbf{y}|\mathbf{z})$ where $\mathbf{X}, \mathbf{Y}, \mathbf{Z}$ are disjoint subsets of $\mathbf{V}$, we have the following.*

$$P_{\mathbf{x}}(\mathbf{y}|\mathbf{z}) = \sum_{\mathbf{d} \setminus \mathbf{y}} Q[\mathbf{D}|\mathbf{Z}] \tag{3}$$

*where $\mathbf{D} = An(\mathbf{Y} \cup \mathbf{Z})_{\mathcal{P}_{\mathbf{V} \setminus \mathbf{x}}} \setminus \mathbf{Z}$.*

**Algorithm 2** Recursive routine to decompose $Q[\mathbf{T}|\mathbf{Z}]$.
___
1: **function** DECOMPOSE($\mathcal{P}$, $\mathbf{T}$, $\mathbf{Z}$)
2:     **if** $\mathbf{T} = \emptyset$ **then return** $\emptyset$
   /* In $\mathcal{P}_{\mathbf{T}\cup\mathbf{Z}}$, let $C^{(\cdot)}$ denote the pc-component of $(\cdot)$ in $\mathcal{P}_{\mathbf{T}\cup\mathbf{Z}}$. */
3:     Initialize $\mathbf{X}$ to an arbitrary node in $\mathbf{T}$
4:     Let $\mathbf{A} = \mathrm{Pa}^*(C^{\mathbf{X}}) \cap \mathrm{Pa}^*(C^{\mathbf{T}\cup\mathbf{Z}\setminus C^{\mathbf{X}}})$
5:     **while** $\mathbf{A} \not\subseteq \mathbf{Z}$ **do**
6:         $\mathbf{X} = \mathbf{X} \cup \mathrm{Ch}^*(\mathbf{A} \cap \mathbf{T})$
7:         $\mathbf{A} = \mathrm{Pa}^*(C^{\mathbf{X}}) \cap \mathrm{Pa}^*(C^{\mathbf{T}\cup\mathbf{Z}\setminus C^{\mathbf{X}}})$
   /* Let $\mathbf{T}_1 = C^{\mathbf{X}} \cap \mathbf{T}$ and $\mathbf{T}_2 = \mathbf{T} \setminus \mathbf{T}_1$ */
8:     **return** $\langle \mathbf{T}_1, \mathcal{R}_{\mathbf{X}} \setminus \mathbf{T}_1 \rangle \cup \mathrm{DECOMPOSE}(\mathcal{P}, \mathbf{T}_2, \mathcal{R}_{\mathbf{T}\cup\mathbf{Z}\setminus C^{\mathbf{x}}} \setminus \mathbf{T}_2)$
___

For example, given the PAG in Figure 2d and query $P_x(y|z_1, z_2)$, we can rewrite the conditional causal effect as $\sum_w Q[Y, W|Z_1, Z_2]$.

The following fact plays a crucial role in the derivation of our algorithm.

**Lemma 1.** *Given a PAG $\mathcal{P}$ over $\mathbf{V}$ and any causal diagram $\mathcal{D}$ in the equivalence class represented by $\mathcal{P}$, suppose $\mathbf{X} \subset \mathbf{A} \subseteq \mathbf{V}$, and let $S^{\mathbf{X}}$ and $C^{\mathbf{X}}$ denote the c-component and pc-component of $\mathbf{X}$ in $\mathcal{D}_{\mathbf{A}}$ and $\mathcal{P}_{\mathbf{A}}$, respectively. Then, for every $Y \in \mathbf{A}$, if $Y \in Pa(S^{\mathbf{X}})$ in $\mathcal{D}_{\mathbf{A}}$, then $Y \in Pa^*(C^{\mathbf{X}})$ in $\mathcal{P}_{\mathbf{A}}$, where $Pa^*(\cdot)$ is (the union of the input set and) the set of possible parents due to directed or partially directed edges ($\rightarrow$, $\circ\!\!\rightarrow$).*

In words, given a PAG $\mathcal{P}$ and any diagram $\mathcal{D}$ in the equivalence class, if a node $Y$ is a parent of the c-component of $\mathbf{X}$ in $\mathcal{D}_{\mathbf{A}}$, then $Y$ must be either in the pc-component of $\mathbf{X}$ in $\mathcal{P}_{\mathbf{A}}$ or a possible parent of the pc-component by a non-circle edge. For example, given the PAG in Figure 2a, $C^X = \{X, Z\}$ and $Y \notin Pa^*(C^X)$, hence $Y \notin Pa(S^X)$ in any causal diagram in the equivalence class. It is easy to see why in this simple example. First, $X$ is not in the pc-component of $Y$ so they are not in the same c-component in any causal diagram, by Proposition 1. If $X$ and $Z$ are in the same c-component in some diagram $\mathcal{D}$ and $Y$ is a parent of $Z$, then we have an (unshielded) collider at $Z$ in $\mathcal{D}$, which would contradict the given PAG. This observation generalizes to more complex cases. Note that the property does not necessarily hold if the input to $Pa(\cdot)$ and $Pa^*(\cdot)$ are arbitrary subsets of $\mathbf{V}$ rather than a c-component and a pc-component.

Next, we derive a sufficient condition for decomposing $Q[\mathbf{T}|\mathbf{Z}]$ into two sub-queries.

**Proposition 5.** *Given a PAG $\mathcal{P}$ over $\mathbf{V}$ and $Q[\mathbf{T}|\mathbf{Z}]$, let $\mathbf{X} \subset \mathbf{T} \cup \mathbf{Z}$. The following decomposition holds if $Pa^*(C^{\mathbf{X}}) \cap Pa^*(C^{\mathbf{T}\cup\mathbf{Z}\setminus C^{\mathbf{X}}}) \subseteq \mathbf{Z}$, where $C^{(\cdot)}$ is the set of nodes in the pc-component of $(\cdot)$ in $\mathcal{P}_{\mathbf{T}\cup\mathbf{Z}}$, $\mathcal{R}_{(\cdot)}$ is with respect to $\mathbf{T} \cup \mathbf{Z}$, $\mathbf{T}_1 = C^{\mathbf{X}} \cap \mathbf{T}$, and $\mathbf{T}_2 = \mathbf{T} \setminus \mathbf{T}_1$.*

$$Q[\mathbf{T}|\mathbf{Z}] = Q[\mathbf{T}_1|\mathcal{R}_{\mathbf{X}} \setminus \mathbf{T}_1] \cdot Q[\mathbf{T}_2|\mathcal{R}_{\mathbf{T}\cup\mathbf{Z}\setminus C^{\mathbf{x}}} \setminus \mathbf{T}_2]$$

For example, given $Q[Y, W|Z_1, Z_2]$ and the PAG in Figure 2d, $C^Y = \{Y, Z_2\}$, $\mathrm{Pa}^*(C^Y) = \{Y, Z_2, Z_1\}$, and $\mathrm{Pa}^*(C^{\{W, Z_1\}}) = \{W, Z_1, Z_2\}$. Hence, $\mathrm{Pa}^*(C^Y) \cap \mathrm{Pa}^*(C^{\{W, Z_1\}}) = \{Z_1, Z_2\}$ and the condition of Prop. 5 is satisfied. So, we have the following decomposition.

$$Q[Y, W|Z_1, Z_2] = Q[Y|Z_2, W] \cdot Q[W|Z_1, Z_2] \tag{4}$$

It is important to note that the condition is based on the pc-component of $\mathbf{X} \subset \mathbf{T} \cup \mathbf{Z}$ while the $Q[\cdot]$ decomposition uses the region of $\mathbf{X}$ (Def. 3). The decomposition would still be valid by using the pc-component instead of the region, but using the region has the advantage of keeping together nodes in the same bucket (i.e., nodes that share circle edges). For instance, using the region allows us to keep $W$ and $Z_2$ together in each sub-query. This will be useful in the final algorithm.

Algorithm 2 decomposes $Q[\mathbf{T}|\mathbf{Z}]$ into a product of sub-queries by applying Prop. 5 recursively. In each iteration, the routine finds a subset $\mathbf{X}$ that satisfies the criterion in the proposition (cf. line 5). The first line checks for a base case where $\mathbf{T} = \emptyset$. For example, given $Q[Y|Z_1, Z_2]$ and the PAG in Figure 2c, the function assigns $\mathbf{X}$ to $\{Y\}$. Since $\mathrm{Pa}^*(C^{\mathbf{X}}) = \{Y, Z_2, Z_1\}$ and $\mathrm{Pa}^*(C^{Z_1}) = \{Z_1, Z_2\}$, their intersection satisfies the criterion. Hence, $Q[Y|Z_1, Z_2] = Q[Y|Z_2] \times$

---

**Algorithm 3** CIDP$(\mathbf{x}, \mathbf{y}, \mathbf{z})$ given PAG $\mathcal{P}$

---

    **Input:** three disjoint sets $\mathbf{X}, \mathbf{Y}, \mathbf{Z} \subset \mathbf{V}$
    **Output:** Expression for $P_\mathbf{x}(\mathbf{y}|\mathbf{z})$ or FAIL

  1: Let $\mathbf{D} = \text{An}(\mathbf{Y} \cup \mathbf{Z})_{\mathcal{P}_{\mathbf{V} \setminus \mathbf{x}}} \setminus \mathbf{Z}$
  2: $P_\mathbf{x}(\mathbf{y}|\mathbf{z}) = \sum_{\mathbf{d} \setminus \mathbf{y}} Q[\mathbf{D}|\mathbf{Z}]$                 ▷ Expand query via Prop. 4
  3: $\mathbf{F} = \text{DECOMPOSE}(\mathcal{P}, \mathbf{D}, \mathbf{Z})$              ▷ $\mathbf{F}$ is a set of pairs $\langle \mathbf{D}_i, \mathbf{Z}_i \rangle$

     /* At this point, $P_\mathbf{x}(\mathbf{y}|\mathbf{z}) = \sum_{\mathbf{d} \setminus \mathbf{y}} \prod_i Q[\mathbf{D}_i|\mathbf{Z}_i] = \prod_i \sum_{\mathbf{d}_i \setminus \mathbf{y}} Q[\mathbf{D}_i|\mathbf{Z}_i]$ */
  4: Let $\mathbf{F}^* = \emptyset$
  5: **for each** $\langle \mathbf{D}_i, \mathbf{Z}_i \rangle \in \mathbf{F}$ **do**
  6:      **if** $\mathbf{D}_i \cap \mathbf{Y} \neq \emptyset$ **then**
  7:          $\mathbf{F}^* = \mathbf{F}^* \cup \text{DO-SEE}(\mathcal{P}, \mathbf{D}_i, \mathbf{Z}_i)$

  8: $P_\mathbf{x}(\mathbf{y}|\mathbf{z}) = \prod_{\{i | \langle \mathbf{D}_i, \mathbf{Z}_i \rangle \in \mathbf{F}^*\}} \sum_{\mathbf{d}_i \setminus \mathbf{y}} \dfrac{\text{IDENTIFY}(\mathbf{D}_i \cup \mathbf{Z}_i, \mathbf{V}, P)}{\sum_{\mathbf{d}_i} \text{IDENTIFY}(\mathbf{D}_i \cup \mathbf{Z}_i, \mathbf{V}, P)}$

  9: **function** DO-SEE$(\mathcal{P}, \mathbf{T}, \mathbf{Z})$
     /* Let $\mathbf{B}$ denote a bucket in $\mathcal{P}$ and $C^{(\cdot)}$ denote the pc-component of $(\cdot)$ in $\mathcal{P}_{\mathbf{T} \cup \mathbf{Z} \cup \color{red}{\mathbf{B}}}$ */
10:      **if** $\exists \mathbf{B} \mid \mathbf{B} \cap (\mathbf{T} \cup \mathbf{Z}) \neq \emptyset \wedge \mathbf{B} \nsubseteq (\mathbf{T} \cup \mathbf{Z})$ **then**
11:          **if** $\text{Pa}^*(C^{\mathbf{B} \setminus (\mathbf{T} \cup \mathbf{Z})}) \cap \mathbf{T} = \emptyset$ **then**
12:              **return** DO-SEE$(\mathcal{P}, \mathbf{T}, \mathbf{Z} \cup \mathbf{B} \setminus \mathbf{T})$
13:          **else**
14:              **throw** FAIL
15:      **return** $\langle \mathbf{T}, \mathbf{Z} \rangle$

---

$Q[\emptyset|Z_1, Z_2]$ where $Q[\emptyset|Z_1, Z_2] = 1$ by definition. The base case accounts for the recursive call DECOMPOSE$(\mathcal{P}, \emptyset, \{Z_1, Z_2\})$ which yields $Q[\emptyset|Z_1, Z_2] = 1$. In general, this step simplifies a target $Q[\cdot]$ and facilitates its computation.

To derive our identification algorithm, we use one more trick. Lemma 2 below provides a sufficient criterion where given $Q[\mathbf{T}|\mathbf{Z}]$ and a causal diagram $\mathcal{D}$, we can move a subset $\mathbf{X}$ from the intervention set $\mathbf{V} \setminus (\mathbf{T} \cup \mathbf{Z})$ to the conditioning set.

**Lemma 2.** *Given causal diagram $\mathcal{D}$ and $Q[\mathbf{T}|\mathbf{Z}]$, let $\mathbf{X} \subseteq \mathbf{V} \setminus (\mathbf{T} \cup \mathbf{Z})$ and let $S^\mathbf{X}$ denote the c-component of $\mathbf{X}$ in $\mathcal{D}_{\mathbf{T} \cup \mathbf{Z} \cup \mathbf{X}}$. If $Pa(S^\mathbf{X}) \cap \mathbf{T} = \emptyset$, then $Q[\mathbf{T}|\mathbf{Z}] = Q[\mathbf{T}|\mathbf{Z} \cup \mathbf{X}]$.*

The following proposition generalizes the result in Lemma 2 to PAGs using the property in Lemma 1.

**Proposition 6.** *Given PAG $\mathcal{P}$ and $Q[\mathbf{T}|\mathbf{Z}]$, let $\mathbf{X} \subseteq \mathbf{V} \setminus (\mathbf{T} \cup \mathbf{Z})$ and let $C^\mathbf{X}$ denote the pc-component of $\mathbf{X}$ in $\mathcal{P}_{\mathbf{T} \cup \mathbf{Z} \cup \mathbf{X}}$. If $Pa^*(C^\mathbf{X}) \cap \mathbf{T} = \emptyset$, then $Q[\mathbf{T}|\mathbf{Z}] = Q[\mathbf{T}|\mathbf{Z} \cup \mathbf{X}]$.*

*Proof.* Let $\mathcal{D}$ be any diagram in the equivalence class represented by $\mathcal{P}$. By Lemma 1, if $\text{Pa}^*(C^\mathbf{X}) \cap \mathbf{T} = \emptyset$ in $\mathcal{P}_{\mathbf{T} \cup \mathbf{Z} \cup \mathbf{X}}$, then $\text{Pa}(S^\mathbf{X}) \cap \mathbf{T} = \emptyset$ in $\mathcal{D}_{\mathbf{T} \cup \mathbf{Z} \cup \mathbf{X}}$. Hence, the proposition follows by Lemma 2 since the equation is valid for all the diagrams in the equivalence class. $\square$

For example, given $Q[Y|Z]$ and the PAG in Figure 2a, $\text{Pa}^*(C^X) \cap \{Y\} = \text{Pa}^*(\{X, Z\}) \cap \{Y\} = \emptyset$, hence $Q[Y|Z] = Q[Y|Z, X]$. Similarly, given the PAG in Fig. 2b, $Q[Y|Z, W] = Q[Y|Z, W, X]$.

Finally, we use the above results to construct Algorithm 3 which identifies conditional causal effects. The algorithm is sound by Theorem 1. It starts by computing set $\mathbf{D}$ then expanding the query accordingly in lines 1-2. Then, **CIDP** calls Alg. 2 which decomposes $Q[\mathbf{D}|\mathbf{Z}]$ to sub-queries as the comment below line 3 elaborates. Lines 4-7 achieve two things. First, we drop every unnecessary query $Q[\mathbf{D}_i|\mathbf{Z}_i]$ where $\mathbf{D}_i \cap \mathbf{Y} = \emptyset$ since $\sum_{\mathbf{d}_i} Q[\mathbf{D}_i|\mathbf{Z}_i] = 1$. For each remaining query $Q[\mathbf{D}_i|\mathbf{Z}_i]$, function DO-SEE$(\cdot, \cdot, \cdot)$ searches recursively for a bucket $\mathbf{B}$ in $\mathcal{P}$ such that a strict subset of $\mathbf{B}$ is in $\mathbf{D}_i \cup \mathbf{Z}_i$, and then tries to apply Prop. 6 to obtain $Q[\mathbf{D}_i|\mathbf{Z}_i \cup \mathbf{B} \setminus \mathbf{D}_i]$. Finally, in line 8, we try to compute the target conditional effect by computing each $Q[\mathbf{D}_i|\mathbf{Z}_i] = \frac{Q[\mathbf{D}_i \cup \mathbf{Z}_i]}{\sum_{\mathbf{d}_i} Q[\mathbf{D}_i \cup \mathbf{Z}_i]}$ and calling IDENTIFY$(\mathbf{D}_i \cup \mathbf{Z}_i, \mathbf{V}, \mathcal{P})$ from Alg. 1. **CIDP** does not identify the target effect if either DO-SEE$(\cdot)$ or IDENTIFY$(\cdot)$ throws a FAIL.

**Theorem 1.** *CIDP (Algorithm 3) is sound.*

*Proof Sketch.* Line 2 follows from Proposition 4. Function DECOMPOSE($\cdot, \cdot, \cdot$) is sound by Proposition 5. The second equivalence in the comment after line 3 is justified by the proof of Prop. 5. Line 6 drops from $\mathbf{F}^*$ every $Q[\mathbf{D}_i|\mathbf{Z}_i]$ where $\mathbf{Y} \cap \mathbf{D}_i = \emptyset$ since $\sum_{\mathbf{d}_i} Q[\mathbf{D}_i|\mathbf{Z}_i] = 1$. The soundness of DO-SEE($\cdot, \cdot, \cdot$) follows from Proposition 6. Finally, line 8 is sound by Definition 5 and the correctness of IDENTIFY($\cdot, \cdot, \cdot$) in (Jaber et al., 2019a). $\qquad\square$

### 4.1 Illustrative Example

Consider the effect $P_x(y|z_1, z_2)$ and the PAG in Figure 2d. We have the following from Eq. 4 and Lines 4-7 of the algorithm. Since $\{Y\}, \{Z_2, W\}$ are buckets in the PAG, DO-SEE($\cdot$) does nothing.

$$P_x(y|z_1, z_2) = \sum_w Q[Y, W|Z_1, Z_2] = Q[Y|Z_2, W] \cdot \sum_w Q[W|Z_1, Z_2] = Q[Y|Z_2, W]$$

Then, we call IDENTIFY($\{Y, Z_2, W\}, \mathbf{V}, P$) to compute $Q[\{Y, Z_2, W\}]$ from $P(\mathbf{V})$. Node $Z_1$ is not in the same pc-component with its only child $Y$ in $\mathcal{P}$. Hence, $Q[\{Y, Z_2, W, X\}]$ is identifiable from $P(\mathbf{V})$ by Proposition 2 using the order $X < \{W, Z_2\} < Z_1 < Y$.

$$Q[\{Y, Z_2, W, X\}] = \frac{P(\mathbf{v})}{P(z_1|x, w, z_2)} \times \sum_{z_1'} P(z_1'|x, w, z_2) = P(x, w, z_2) \cdot P(y|x, w, z_2, z_1) \quad (5)$$

Next, $X$ does not have any possible children in $\mathcal{P}_{\mathbf{V}\setminus\{Z_1\}}$, hence $Q[\{Y, Z_2, W\}]$ is identifiable from $Q[\{Y, Z_2, W, X\}]$ (Eq. 5) using the partial order $X < \{W, Z_2\} < Y$. The last equivalence is a simplification obtained by considering the independence relation $(X \perp\!\!\!\perp \{W, Z_2\})$.

$$Q[\{Y, Z_2, W\}] = \frac{P_{z_1}}{P_{z_1}(x)} \times \sum_{x'} P_{z_1}(x') = \frac{P(x, w, z_2) P(y|x, w, z_2, z_1)}{P(x)} = P(w, z_2) P(y|w, z_2, z_1)$$

Finally, the conditional effect simplifies as follows.

$$P_x(y|z_1, z_2) = \frac{Q[\{Y, Z_2, W\}]}{\sum_y Q[\{Y, Z_2, W\}]} = \frac{P(w, z_2) \cdot P(y|w, z_2, z_1)}{\sum_y P(w, z_2) \cdot P(y|w, z_2, z_1)} = P(y|z_2, z_1)$$

### 4.2 Expressiveness

Theorem 2 below establishes that **CIDP** subsumes **IDP**. Conversely, **IDP** cannot compute some conditional effects that are identifiable by **CIDP**, such as the cases depicted in Fig. 2, because the corresponding joint effects are not identifiable. Hence, **CIDP** is strictly more powerful than **IDP**.

**Theorem 2.** *CIDP (Alg. 3) subsumes IDP (Alg. 1) $-$ if CIDP fails to identify $P_{\mathbf{x}}(\mathbf{y})$, IDP fails too.*

*Proof.* Suppose $\mathbf{Z} = \emptyset$. The query expansion reduces to that in Alg. 1. Alg. 2 will decompose $Q[\mathbf{D}]$ only if the subsets are disjoint in $\mathcal{P}_{\mathbf{D}}$ since any adjacency implies the condition of Proposition 5 would fail. Such a decomposition is valid for **IDP** using Prop. 3 where the denominator set would be empty. Whenever set $\mathbf{B}$ in DO-SEE($\cdot$) exists (line 10), the function fails. We then have $\mathbf{B} \cap \mathbf{X} \neq \emptyset$ and there exists a potentially causal path from $\mathbf{X}$ to $\mathbf{Y}$ that starts with an invisible edge. Hence, $P_{\mathbf{x}}(\mathbf{y})$ is not identifiable by (Jaber et al., 2019a, Th. 3), and consequently **IDP** fails. Finally, **CIDP** fails if a call to IDENTIFY($\cdot$) fails. It follows that **IDP** would fail as well which concludes the proof. $\qquad\square$

## 5 Conclusion

The problem of identifying conditional causal effects is of great interest due to its role in evaluating conditional plans or policies (Pearl and Robins, 1995). We have investigated a challenging version of this problem where in addition to the observational distribution, the available causal information is not a fully specified causal diagram, but a PAG which represents a Markov equivalence class of causal diagrams and which can be inferred from the observational distribution. We develop an algorithm to compute the effect of an arbitrary set of intervention variables $\mathbf{X}$ on an arbitrary outcome set $\mathbf{Y}$ while conditioning on a third disjoint set $\mathbf{Z}$, denoted $P_{\mathbf{x}}(\mathbf{y}|\mathbf{z})$. We show that the proposed algorithm subsumes the state-of-the-art algorithm in (Jaber et al., 2019a), which is complete for unconditional effects. Moreover, **CIDP** identifies all the examples in the literature that we are aware of, including the one in Fig. 2b which is not identifiable by the generalized do-calculus (Zhang, 2008a). Based on these observations, we conjecture that our algorithm is complete.

## Acknowledgements

Bareinboim and Jaber are supported in parts by grants from NSF IIS-1704352, IIS-1750807 (CA-REER), IBM Research, and Adobe Research. Zhang's research was supported in part by the Research Grants Council of Hong Kong under the General Research Fund LU13602818.

## Footnotes

[1] Another approach is based on SAT (Boolean constraint satisfaction) solvers (Hyttinen et al., 2015). Given its somewhat distinct nature, a closer comparison lies outside the scope of this paper.

[2]The proofs can be found in (Jaber et al., 2019b).

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
