[Reviews · NeurIPS 2019]

Reviewer 1



The research is carried out in the problem of identification of causal effects in the frame of probabilistic causality established by Pearl, Spirtes, Glymour and Scheines. The current manuscript expands from current work on unconditional causal effects to conditional causal effects. The main rationale of the solution (alike a previous related one) is to reach the computation of the post-interventional causal effect Q using in the denominator a summatory over the elements of a partition of the causal graph, as long as such partition complies with some condition. As from here, my understanding of the paper I must accept was very limited. The rationale of the idea seems ok, but I’ve been unable to follow the details of the maths (this is sharply different from not having check them! I simply did not fully understood them despite making several attempts at understanding them!). In this sense, I can only apologize to the authors for not having been able to give a more thorough feedback (I have lowered my confidence to the minimum). STRONG POINTS + Given my limited understanding, but as far as I can confirm, the demonstrations of the propositions and lemmas are correct. This is concentrated in the supplementary material.• WEAK POINTS + To help contextualizing, the draft contains a lot of material already available in other papers (likely from the same authors) leaving little room for the new material. In fact, most of the real innovation is what it is given in the supplementary material. Although I found this useful in a really difficult topic, but to an experienced reader (not me!), it may left him with a bit of a deja-vu impression.

Reviewer 2



Originality; in my humble opinion the paper is moderately original but it addresses and solves a relevant and difficult problem of inference and decision making under partial knowledge and when only observational data are made available. Quality; the technical quality of the paper is excellent, it is well written and structured. However, it is extremely difficult to review a paper like this one in so few days (we have to keep in mind that 5 papers to review in 3 weeks is really an incredible overhead in the case where technical papers like this one have to be judged. Clarity; the paper is well structured and reads well. However, I would stress that is more examples could be inserted to present and discuss the paper this will help the interested reader to grab the many innovations and contributions of the paper. Significance; the paper is highly significant. Indeed, the developed theory and the designed algorithm will help to analyze complex situations, i.e. real world situations where the available knowledge and data are limited and in particular they are of observational nature. I also have few questions for the authors/s; Pag. 2; line five from Structural Causal Models, it is assumed that no U elements can be can be parents of V? Pag. 2; would you please clarify what small pai means in formula (1), is it the specific value of Pai? If so then I do not understand well formula (1), could you please help me understand it?

Reviewer 3



The paper focuses on an interesting and useful problem (significance), and it provides a novel theoretical contribution (originality) that seems technically correct (high quality). As mentioned in the previous section, it extends the previous work from (Jaber et al. 2019) in terms of conditional causal effects, providing an algorithm that is sound and conjectured to be complete (the lack of this proof is possibly one of the few negative points of the paper). In terms of clarity, the paper seems quite well-written, although I do wonder if a beginner in causal inference would be able to read it. Minor details and typos: L65: not sure about the “semantical” framework, possibly rephrase L121-126: could be improved in terms of clarity, possibly with an example? L272: there existS a potentially causal path

[Author Response · NeurIPS 2019]

We thank the reviewers for spending their time and providing such a favorable assessment and helpful feedback. We
will try to incorporate the suggestions into our manuscript. Below, we address the main issues raised by the reviewers.

**Reviewer 1:**

– Background material: Our aim was to make the paper as self-contained as possible, however, we do agree that it
would be desirable to have more space to expound the new results. There is a non-trivial trade-off but we will try to
shorten the background material and provide further elaboration and examples for the new results.

– Related work: We described the state of the art in the introduction. The problem of identifying conditional causal
effects given a fully specified causal diagram has been solved by [Tian, 2004; Shpitser and Pearl, 2006], which
presented complete algorithms. In this paper, we are dealing with the problem of conditional causal identification
given an equivalence class of causal diagrams represented by a PAG, rather than a single causal diagram. This
problem setting is more realistic, but it's also significantly more challenging, which we suspect is one of the reasons
relatively fewer results are available in the literature. Among the attempts to solve this problem are [Zhang, 2007]
and [Jaber et al., 2019], but each had its shortcomings as discussed in the introduction (lines 45-53) and Section 4
(lines 168-176), which indeed motivated this work.

– "The $do(X = x)$ operation require replacing the value $x$ for $X$ but also removing all incoming influences on node
$X$.": Your observation is correct. Note that "replacing the original equation for $X$ by the constant $x$" does eliminate
all incoming influence on $X$ as we are modifying the pre-interventional (natural) structural function underlying $X$
(or $f_x$), and not just conditioning on the constant value $x$.

– Definite c-component: Throughout the paper, we use the abbreviation "dc-component" for "definite c-component",
which is defined on pg. 4, lines 132-133. We will make this clearer and double check other such terms.

**Reviewer 2:**

– Set $\mathbf{U}$ in SCM definition: We apologize for the confusion. Actually variables in $\mathbf{U}$ can only be parents of variables in
$\mathbf{V}$. For the purposes of this paper, it is helpful to distinguish between observed parents of a variable $V_i$, which are
denoted as $\mathbf{Pa}_i$ (and is a subset of $\mathbf{V}$), and unobserved parents of $V_i$, which are denoted as $\mathbf{U}_i$ (and is a subset of $\mathbf{U}$).

– Small $\mathbf{pa}_i$: This is indeed a specific value assignment to the variables set $\mathbf{Pa}_i$.

– Formula 1: To illustrate the formula, consider an SCM where $\mathbf{V} = \{V_1, V_2\}$, $\mathbf{U} = \{U\}$, $\mathbf{F} = \{V_1 \leftarrow f_1(U), V_2 \leftarrow f_2(V_1, U)\}$, and $P(U)$. Thus, we have $\mathbf{Pa}_1 = \emptyset$, $\mathbf{Pa}_2 = \{V_1\}$, $\mathbf{U}_1 = \{U\}$, and $\mathbf{U}_2 = \{U\}$. The corresponding causal diagram is $G = \{V_1 \rightarrow V_2, V_1 \leftrightarrow V_2\}$, where $V_1 \leftrightarrow V_2$ represents the unmeasured common cause/parent $U$ of $V_1$ and $V_2$. Any distribution generated by the above model factorizes as follows:

$$P(V_1, V_2, U) = P(V_1|U)P(V_2|V_1, U)P(U)$$

Recall that $U$ is an unmeasured variable, thus what we can sample from is the marginal distribution over $\{V_1, V_2\}$ which can be written as follows.

$$P(\mathbf{V}) = \sum_u P(V_1, V_2, u) = \sum_u P(V_1|u)P(V_2|V_1, u)P(u)$$

The formula can be written with lower case letters standing for the value assignments of the variables, which leads to
a formula akin to the one in the paper.

– Clarity/more examples: We will add more examples in the paper to improve the readability and clarity.

– Code: We certainly agree with your suggestion and plan to make the code of our work available, which we expect
can lead to a smoother connection between the theory and its practical use. We appreciate your suggestion.

**Reviewer 3:**

– Completeness: We share the "feeling" of the reviewer regarding the completeness proof, but we do not yet have it yet.
Having said that, we also would like to share that we strongly believe the conjecture is true and we are working hard
towards establishing this statement more formally.

– Clarity: We tried to make the paper as self-contained as possible within the page limit, but we understand that a more
thorough discussion would be more readable for the wider audience. We will do our best to improve the clarity of the
paper for beginners in causal inference. Also, since the focus of the paper is more on the identifiability part, and not
yet on estimation from finite samples, we didn't include numerical examples/empirical results. Still, we plan to try to
account for this challenge in the near future.

[Meta-Review · NeurIPS 2019]

In this paper, the authors consider the problem of identifying conditional causal effects from an equivalence class of MAGs. The paper was well received, and appears to be more involved than usual conditional causal effect identification results. The only issue is lack of completeness.